# The Neglected Tropical Disease Non-Governmental Organization Network: The role of a global health network in an evolving global health landscape

**Girija Sankar** 🄲¹*, **Arielle Dolegui²**, **Jemish Acharya³**

**1** Inclusive Health Initiative, Christian Blind Mission, CBM, Bensheim, Germany, **2** Gates Foundation, Washington, DC, United States of America, **3** The Leprosy Mission Nepal, Kathmandu, Nepal

* girija.sankar@cbm.org

## Abstract

The Neglected Tropical Diseases NGO Network (NNN) was established in 2009 to bring together non-governmental organizations (NGOs) working towards the control and elimination of neglected tropical diseases (NTDs). In response to the evolving landscape of global health and the increasing focus on equity, the NNN conducted a membership analysis in 2021. This analysis aimed to assess the network's composition and connectivity, revealing significant insights into its current structure and areas for improvement. Key findings from the analysis include the following: a majority of NNN members were based in high-income countries (HICs); 40% of the member organizations concentrated on a single NTD; the African region had nearly twice the number of member organizations supporting interventions in the region compared to other regions; and, members from HICs tend to be better networked within the NNN. These findings highlight the need for the NNN to enhance its inclusivity and representation, ensuring that voices from endemic countries are more prominently integrated into the network. By aligning with the NTD Road Map 2021–2030, the NNN aims to address these disparities and strengthen its commitment to global health equity. This analysis serves as a call to action for other global health networks to undertake similar self-assessments. Through transparency and critical reflection, the pursuit of global health equity can be advanced, ensuring that all regions and diseases receive appropriate attention and resources.

## Introduction

The Neglected Tropical Diseases Non-Governmental Organization Network or the NNN is a global forum for NGOs to contribute to the control and elimination of NTDs outlined in the World Health Organization (WHO) NTD road map [1]. It was founded in 2009 by the onchocerciasis, lymphatic filariasis and trachoma disease knowledge communities/coalitions at a meeting in Accra, Ghana [2]. The soil-transmitted worm infection and schistosomiasis disease communities joined the NNN soon after, followed by the leprosy coalitions in 2013. Initially intended as a common platform for the disease-specific communities of practice,

**Data availability statement:** All relevant data are within the paper and its Supporting information files.

**Funding:** The authors received no specific funding for this work.

**Competing interests:** The authors have declared that no competing interests exist.

the NNN has since grown to convening practitioners on priority and agenda-setting related to cross-cutting topics such as water, sanitation, and hygiene (WASH), One Health, sustainability, health systems strengthening, and serving communities in conflict and humanitarian emergencies.

The annual conference organized by the NNN is a key global event that brings a diverse group of stakeholders together in advancing NTD elimination priorities around the world [3,4].

What started as a small community of implementing partner organizations mostly based in the Global North is now a forum of 97 public health, development, and humanitarian non-profit organizations, academic institutions, research centers, product development organizations, and other disease-specific, or geographic alliances [5] (At the time of analysis, the NNN had 91 members; however, as of September 2023, the membership grew to 97). NNN members have contributed to several key policy statements and guidelines such as the Behavior, Environment, Social Inclusion, and Treatment and care (BEST) framework in 2016, the WHO and NNN joint toolkit on water, sanitation, and hygiene (WASH) and NTDs in 2018, the NNN commitment on patient safety and statement on sustainability in 2019 and the NNN statement of commitment to the participation of people affected by NTDs in 2020 [6,7].

## NNN structure and governance

The NNN has a two-tier governance structure that includes an Executive and Steering Committee, with working groups, task teams and task groups organized by disease and theme that produce work products serving the network and the broader NTD community (Fig 1).

The NNN has an annual election cycle to elect a Vice-Chair who serves in the role for 12 months, and steps into the role of Chair at the following annual conference. The Chair, Vice-Chair, and the Immediate Past Chair together form the NNN Steering Committee which provides strategic and tactical oversight for the NNN activities. In addition to the Steering Committee, the chairs and co-chairs of the working groups form the Executive Committee which meets at least once a quarter. The NNN annual conference is the network's marquee event that brings together members but is also open to the NTD community-at-large. The leadership roles within the NNN governance are 100% volunteer driven. Unlike other disease coalitions, the governance, and day-to-day operations of the NNN are also on volunteer time.

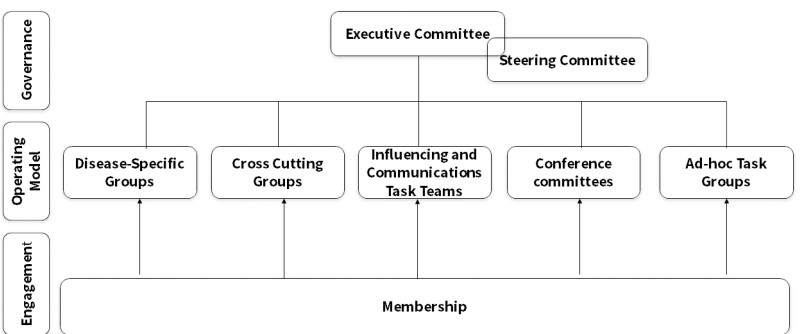

**Fig 1. Structure of the NNN source:** [8].

### Need for critical reflection

The geopolitical and political economic landscape of neglected tropical diseases (NTDs) has evolved significantly since the network's founding. In January 2012, the London Declaration united pharmaceutical companies, donors, endemic countries, and NGOs to control, eliminate, or eradicate 10 NTDs by 2020, improving over a billion lives. This declaration boosted global funding and political attention, eventually leading to the elimination of at least one NTD in fifty countries and increased visibility and domestic financing for NTD programmes [1,9–13]. Two global roadmaps for NTDs were developed in 2012 and 2021, emphasizing country ownership and the inclusion of NTDs in Sustainable Development Goals [1,14–16].

The NNN was established by a coming together of NTD practitioners drawn from NGOs that have played a critical role in the design, scale-up, and success of large scale NTD programmes around the world [17–19]. NGOs have been instrumental in managing public-private partnerships responsible for some of the longest-running pharmaceutical donations in global health [20–23]. However, that the NNN was established by practitioners and organizations primarily from high-income countries was an uncomfortable reality facing the volunteer executive leadership of the NNN. This prompted a collective reflection within the network's leadership on the evolving landscape within which the NNN operates, and the need to assess the network's operating structures and governance to prepare the ground for a more equitable, inclusive, and diverse network.

The COVID-19 pandemic further highlighted health inequities and inspired global health reflections [24–28]. This inspired a review of the NNN's mission, vision, and governance to align with the needs of endemic countries and affected communities. In preparation for its 12th annual conference in September 2021, the NNN conducted a membership analysis to support this shift and respond to calls for decolonizing global health and promoting country ownership in NTD programmes.

The need to review the NNN membership and structure was also in response to the twin salvos issued by the broader global health and NTD community – to decolonize global health, and to promote greater country ownership in NTD programmes.

The objective of the membership analysis was to:

1) Analyze and classify membership composition: assess the NNN's membership by organizational size, disease focus, and geographical distribution.

2) Review member engagement: evaluate the extent of members' participation within the network through representation in the network's working groups.

3) Assess member cohesion: assess the extent to which members are connected with each other through organizational affiliations with other alliances, associations, or networks.

## Methods

An analysis of the current membership of the Neglected Tropical Diseases (NTDs) NGO Network (NNN) was conducted to better understand the composition of the NNN by geographical coverage, disease focus, organizational size, extent of network participation, and inter-connectivity with other member organizations.

### Data collection

Data for the membership analysis was sourced from the membership list maintained by the NNN steering committee. This list, reflective of members as of July 2021, included the names and contact information of all member organizations.

### Desk review

A desk review was conducted using organizational websites, annual reports, and other grey literature. Each member organization was classified by the following criteria:

Type: Health NGO, humanitarian NGO, global development NGO, academic institution, or cross-sectoral alliance.

Location of headquarters: The country where the organization's main office is located.

Focus of NTD activities: Specific diseases targeted, and types of activities conducted (e.g., treatment, prevention, research).

Implementation regions: Geographical regions where the organization operates.

Participation in other global alliances, networks, or associations: Membership in other relevant networks or coalitions.

### Data analysis

The data were analyzed to classify the membership composition by geographical distribution, organizational type, and disease focus. Specific metrics included:

Geographical distribution: Number and percentage of organizations headquartered in high-income versus low- and middle-income countries, further broken down by specific countries.

Organizational type: Proportion of different types of organizations.

Disease focus: Percentage of organizations focusing on specific NTDs and the range of activities undertaken.

Inter-connectivity and network participation: The extent of network participation and inter-connectivity among member organizations was assessed by:

- Identifying shared memberships in other global health networks.
- Analyzing collaborative projects or initiatives reported in annual reports or organizational websites.

## Results

The results are organized by the objectives.

1)  Analyze and classify membership composition: assess the NNN's membership by geographical distribution, organizational type, and disease focus.

At the time of analysis, the NNN had 91 member organizations. Of these, 74 or 81% were headquartered in the high-income countries and 17, or 19% in lower-middle or low-income countries [29]. Notably, 40% of the 74 member organizations that were based in high-income countries were headquartered in the USA, and 24% in the UK (Table 1).

Sixty-nine or nearly 76% of the members were non-profit, public health, humanitarian, global development organizations; 15% were coalitions, alliances, or associations, and the remaining nine percent were academic or research institutions. Coalitions in this context are formal or informal alliances or groupings of organizations pursuing similar priorities and include alliances such as the International Coalition for Trachoma Control (ICTC), the International Federation of Anti-Leprosy Associations (ILEP), and the Global Schistosomiasis Alliance (GSA) [2,30,31].

With respect to disease focus, 55 members or 60% focused on one or more NTD, and the rest, 40%, on only one disease. Forty-four percent of the members focused on diseases targeted

for elimination as a public health problem, and 31% focused on programme implementation for diseases targeted for interruption of transmission (some members may have been counted more than once depending on the disease groups they support) (Fig 2).

We used the WHO regional classification system to examine the countries where NNN members supported NTD interventions – 73% of the member organizations (67 out of 91) supported NTD interventions in the African region. Other regions in order of member focus included the Southeast Asian region, the Americas, Western Pacific, and the Eastern Mediterranean region. The geographic focus of 12 members could not be appropriately determined.

**Table 1.  NNN membership by country.**

| High income | | Low, lower-middle-income | |
|---|---|---|---|
| **Country** | **Number of members** | **Country** | **Number of members** |
| Argentina | 1 | DRC | 1 |
| Australia | 5 | Ethiopia | 2 |
| Austria | 2 | Ghana | 1 |
| Belgium | 1 | India | 2 |
| Canada | 2 | Kenya | 2 |
| France | 1 | Madagascar | 1 |
| Germany | 5 | Nigeria | 3 |
| Japan | 1 | Senegal | 1 |
| Netherlands | 1 | Somalia | 1 |
| South Korea | 1 | South Africa | 1 |
| Spain | 2 | Tanzania | 1 |
| Switzerland | 3 | Yemen | 1 |
| UAE | 1 | – | – |
| UK | 18 | – | – |
| USA | 30 | – | – |
| **Total** | **74** | **Total** | **17** |

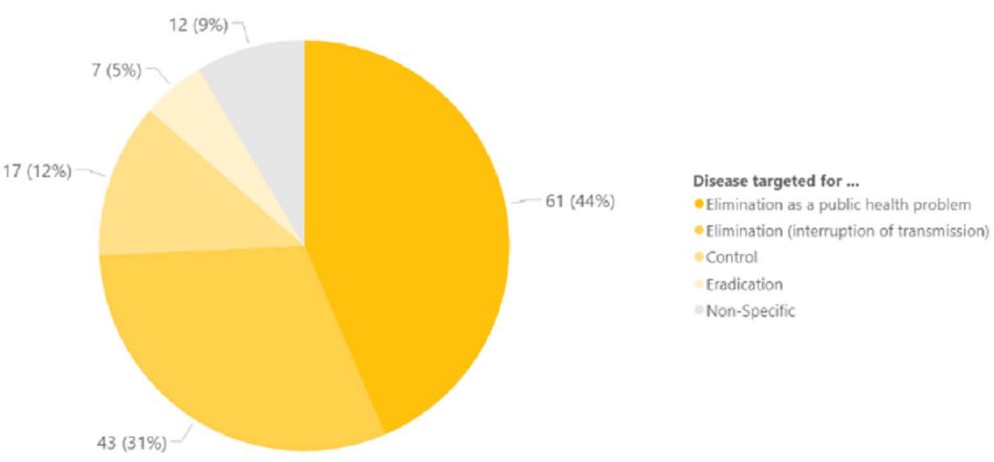

**Fig 2.  Disease focus of NNN members.**

While countries in Africa bear the highest burden of some diseases (i.e., trachoma in Ethiopia or onchocerciasis in the Democratic Republic of the Congo), collectively, and based on the most recent WHO global report on NTDs 2024, 51.4% of the 1.62 billion people requiring interventions for NTDs were in the WHO South-East Asia Region, followed by the African Region at 35.8% [13]. This finding may point to the need for increased engagement with potential or new members active in NTD elimination in the South-East Asia Region.

2) Review member engagement: evaluate the extent of members' participation within the network.

Next, we analyzed the membership and participation across the communities of practice or working groups within the NNN. Briefly, the NNN hosts several disease-specific and cross-sectoral or cross-cutting groups that serve as communities of practice primarily for employees of NNN member organizations but also open to other individuals not affiliated to member organizations but otherwise interested in NTDs. Of the 91 members, staff from 35 organizations participated in one community of practice. Of the 35, six or 17% were from middle or low-income countries, and 83% or the majority, from high-income countries. Twenty-two member organizations, all from high-income countries, had staff participating in two or more communities of practice. One organization, based in a high-income country, had staff members participating in four communities of practice or working groups (Fig 3).

3) Assess member cohesion: assess the extent to which members are connected with each other through organizational affiliations with other alliances, associations, or networks.

Finally, we examined how and if members were connected to each other beyond their common affiliation to the NNN. A network pathways analysis showed that members from high-income countries were more connected with other members, primarily through coalitions, alliances, or associations, than members from middle- or low-income countries. Only two members from low or middle-income countries were connected to other coalitions (Fig 4).

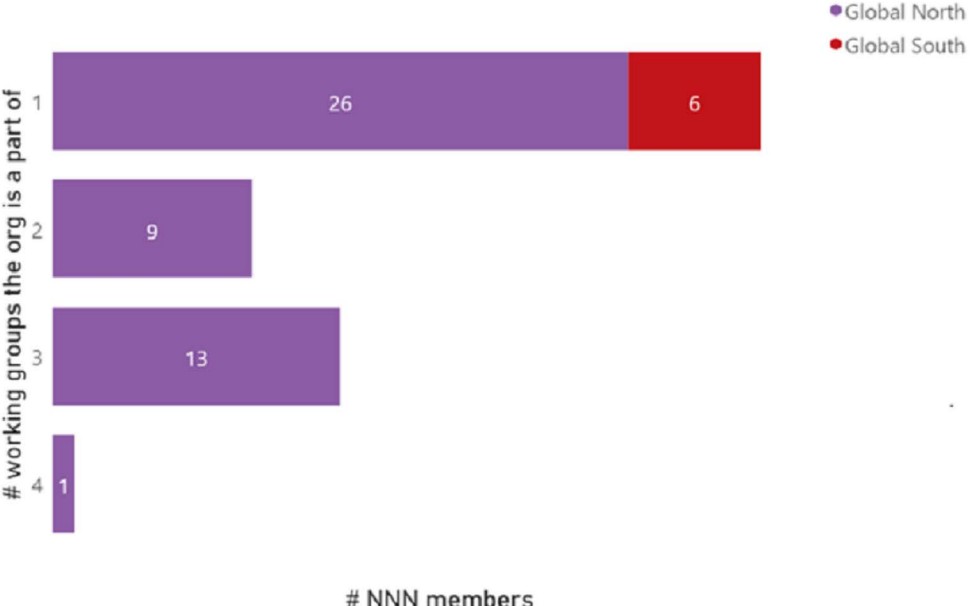

**Fig 3. NNN membership participation in working groups.**

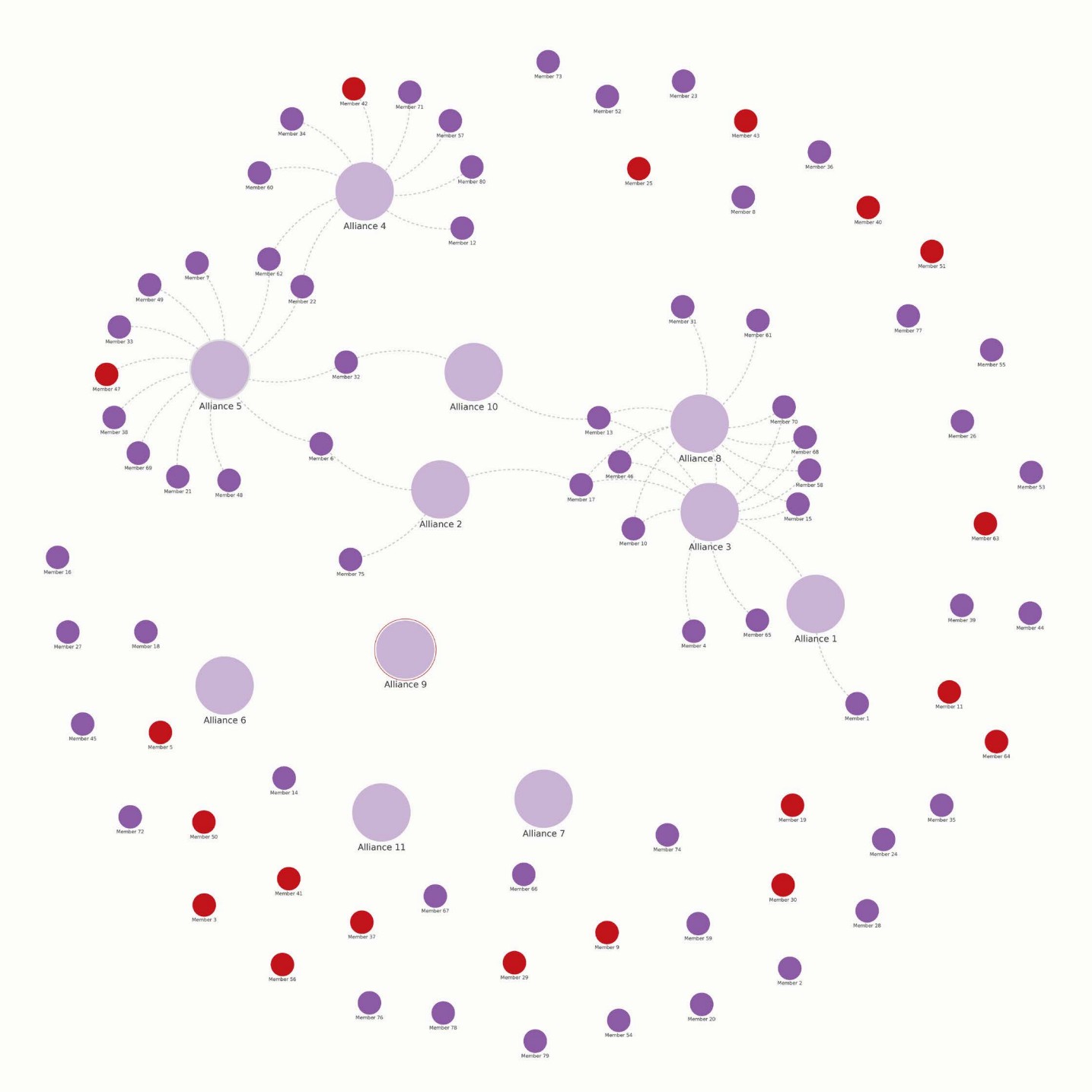

**Fig 4. Network pathways.** Large purple circles-alliances or coalitions that are members of the NNN; small purple circles – NNN members from the Global North; small red circles – NNN members from the Global South.

To summarize, most members were from high-income countries (HICs), 40% focused on only one NTD, the African region had approximately two times as many members supporting interventions in the region as compared to any other region, and some of the members (from HICs) were better networked than most members.

## Discussion

The analysis was also a response to the structural inequities laid bare by the COVID-19 pandemic and the increasing calls to re-examine global health practices by the burgeoning "decolonizing global health" movement [24,27,28,32–35].

These global events precipitated introspection and self-reflection at the global and organizational levels and one that the NNN also embraced in undertaking a strategic review of its role, its operating structures, membership, and lay the foundations for a network that is responsive to the changing NTD and global health governance principles [24,27,28,36–38].

In reflecting on the changing global landscape, this self-reflective exercise revealed structural inequities in our very own network: The NNN is a global network of NGO and yet, only 19% of our members are from the Global South and countries where we support NTD interventions. Additionally, all but one of the members of the NNN's steering committee have come from NGO members in high-income countries. This is more a reflection of the state of most global coalitions and networks that support public health interventions in the Global South [39,40].

When the NNN was established in 2009, the NTD and global health landscape was vastly different from what it is today. However, as Plamondon et al argue, global health partnerships, "should do something to advance equity action, regardless of whether or not the partnership was established with that intention" [41]. One of the NNN's external facing objectives is "to be the unified voice on common issues to achieve our NTD goals," [8]. The membership analysis has shown the NNN has not lived up to this objective as there is an unbalanced representation within the network, with its members predominantly from the Global North. The phrase, "*unified voice*" evades and disguises the politics of north-south and donor-recipient country differences. Implicit in the framing of the objective in serving as the "unified voice" is the boundlessness of the geographies, that all members are somehow equal.

What does it mean to be a unifying voice in a landscape of 91 members, some of whom are global entities with multi-million dollar budgets, present in over 20 countries and some of whom are small, single issue, single geography organizations [42]? Given the inherent limitations of these objectives, we instead undertook an analysis of what our coalition is today as far as representation, geography, participation, and compare that against our values and objectives to see if these objectives require a review and readjustment.

The founding principles and values of the NNN or any such similar network, under girded by values such as unifying voice, and serving as a common platform do not lend themselves easily to evaluations, not least because the NNN was and remains a volunteer, informal, network. But a membership analysis provides a reasonable starting point for a more critical reflection on the purpose and place of such a network in a changing landscape.

What is unstated until now in this review is the idea of equity, which is arguably the fundamental question under girding this exercise – is the NNN an equitable platform for NGOs? Are its structures, governance, processes, work outputs fair, and equitable?

Drawing on Plamondon et al.'s work, the purpose of this analysis was not as a commentary on the effectiveness or efficiency of the NNN but as a starting point to a shared and improved understanding of how the process of networking itself could be responsive to issues of equity and power [41].

## Limitations

The analysis and discussion presented in this paper were limited to the data found on member websites and promotional material. The data analyzed was based on what has been presented to the NNN by the members. Future research could supplement this analysis with key informant interviews and surveys of members to explore the perceived values of the NNN, and its evolution towards more inclusion and diversity.

The analysis of the interconnectedness of the members was based on publicly available information of member participation in other coalitions or alliances which may not be fully reflective of the members' own perceptions of their levels of participation in these other coalitions.

The analysis is also limited in its scope, with a focus almost entirely on the geographical distribution of the NNN members into the two broad categories of Global North (high-income countries) and Global South (lower-middle or low-income countries).

## Conclusion

While this analysis has its limitations, as a starting point to further critical reflection, it offers initial recommendations for the NNN and similar networks to consider:

1) Grow the membership base to include more community-based, civil-society, organizations that are local, and close to the communities being served by the NTD programmes. If networks such as the NNN are to remain true to their founding principles and values of serving as a global and common platform uniting individuals and organizations under a common cause, the globality of such a value can only be demonstrated by "strength in numbers," i.e., greater representation, participation and ownership across a spectrum of organization types.

2) Reduce the barriers to entry for active participation in group activities. The membership analysis hinted at several factors that may be affecting the proactive engagement of organizations from the Global South in network activities, such as participation in working groups. A further review might elucidate the reasons behind less engagement from members in the Global South and provide insights on how best to engage them in network. Such factors may include: language barriers, insufficient staff time available to devote to external representation and participation, perceptions around evidence (or lack thereof) of benefits from participating in network activities, and perceptions around inherent power dynamics between members [43–45].

3) Ground the network in the everyday realities of the communities who are at risk, the civic groups, associations, and organizations who represent affected, and at-risk communities, the policy makers and public sector practitioners who legislate and execute public policy that affects the lives of at-risk and affected communities.

4) Beyond structural considerations, an implication of this analysis is the importance of building efficiencies and synergies within NTD programmes. As part of this evolution, the NNN could explore co-delivery models within its subgroups, where interventions for multiple NTDs are combined in a single campaign targeting the same co-endemic communities. By collaborating on delivery channels and establishing shared metrics reflecting disease-specific outcomes and cross-disease health improvements, NNN members could provide a clearer impact narrative to donors and stakeholders.

5) The need for further critical reflection and evaluation of the benefits of networks such as the NNN by engaging with communities affected by NTDs, national, and regional policy makers whose "ownership" we seek to increase in NTD programmes [1]. This initial

reflection on the NNN should serve as a starting point to engage with more difficult, complex, and ethical challenges around the continued relevance of "global" communities of practices to local public health challenges.

As Michael and Madon argue, "the cultivation of "epistemic" networks of practitioners that share a common philosophical and theoretical mental construct in policy making also leads to the discipline-bounded vertical solutions typically observed in global health delivery," [46]. In effect, these specialized knowledge communities have led to "hyper-specialization" of expertise and praxis that favor experts and practitioners from the Global North who have access to resources, job opportunities, and specialized training for such roles. Following on from this critical commentary, a question to ask in further critical reflection would be: "is the NNN perpetuating or ameliorating power imbalances?"

The NNN membership analysis was instigated in part by the very successes of the NTD community because with success comes complacency, entrenched power, authority and influence, and a reticence to challenge status quo [44].

The analysis presented in this paper is a starting point for further reflection and introspection on how we do what we do in global health. A redistribution of the geographical representation of the NNN's membership alone may not be enough to align the network with the current global landscape. Boosting membership from the Global South or from countries implementing NTD programmes would be a critical next step but that alone is not sufficient to promote equity in global health networks like the NNN.

Decolonizing NTD practice surely does not just end with inviting more "local" or "endemic country" representation in the network, but it's a starting point, that should lead to more meaningful engagement over time [47].

The global health landscape is replete with communities of practice, alliances, coalitions, networks, each one feeding off of the other, some in transparent ways, and some not – between their titular, nominal presence in some groups, and active participation in others, NGOs in high-income countries or the Global North have myriad ways of accessing and sharing knowledge, connecting with potential donors, partners, and gaining access to influencers and decision-makers. The location of many of these alliances in high-income countries also make it less accessible to NGOs in the Global South. For the NNN to deliver on the vision of the new road map towards country ownership, global health partnerships such as the NNN must review their membership, governance, and processes periodically to ensure that their mandate can indeed deliver on the goals of serving as the unified NGO voice on common issues to achieve NTD goals.

## Supporting information

**S1 Data. NNN membership.**
(XLSX)

## Acknowledgments

The authors are grateful to Suzanne Fritzsche and Paryhse May for supporting data collection and analysis. Many thanks also to Aparna Barua Adams, Emma Harding-Esch, and Yael Velleman for insightful feedback on earlier versions of this work presented at the NTD NGO Annual Conference, September 2021.

## Author contributions

**Conceptualization:** Girija Sankar, Arielle Dolegui.

**Data curation:** Girija Sankar.

**Formal analysis:** Girija Sankar, Arielle Dolegui.

**Project administration:** Girija Sankar.

**Visualization:** Girija Sankar.

**Writing – original draft:** Girija Sankar.

**Writing – review & editing:** Girija Sankar, Arielle Dolegui, Jemish Acharya.

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
