## [Decision Letter · Decision Letter 0]

9 Oct 2024

PGPH-D-24-01554

The neglected tropical disease non-governmental organization network: the role of a global health network in an evolving global health landscape

Dear Dr. Sankar,

Thank you for submitting your manuscript to PLOS Global Public Health. After careful consideration, we feel that it has merit but does not fully meet PLOS Global Public Health’s publication criteria as it currently stands. Therefore, we invite you to submit a revised version of the manuscript that addresses the points raised during the review process.

Please address the comments of Reviewer 1 and Reviewer 2 in full, and explain your decisions and changes. Additionally, re: the point regarding the African region receiving twice the support compared to other regions. In addition to the discussion requested by Reviewer 2, I ask that you reflect on the question of denominator here. I.e. If Africa were to have twice the burden of NTDs compared to the rest of the world, then this would arguably be equitable and warranted, if they have more than twice the NTDs than the rest of the world, this may actually reflect inadequate support, etc. Please do your best to address this point. 

We look forward to receiving your revised manuscript.

Kind regards,

Bram Wispelwey, MD, MS, MPH

Academic Editor

Journal Requirements:

1. We ask that a manuscript source file is provided at Revision. Please upload your manuscript file as a .doc, .docx, .rtf or .tex.

Reviewers' comments:

Reviewer's Responses to Questions

**Comments to the Author**

1. Does this manuscript meet PLOS Global Public Health’s publication criteria ? Is the manuscript technically sound, and do the data support the conclusions? The manuscript must describe methodologically and ethically rigorous research with conclusions that are appropriately drawn based on the data presented.

Reviewer #1: Yes

Reviewer #2: Yes

2. Has the statistical analysis been performed appropriately and rigorously?

Reviewer #1: Yes

Reviewer #2: I don't know

3. Have the authors made all data underlying the findings in their manuscript fully available (please refer to the Data Availability Statement at the start of the manuscript PDF file)?

Reviewer #1: Yes

Reviewer #2: Yes

4. Is the manuscript presented in an intelligible fashion and written in standard English?

Reviewer #1: Yes

Reviewer #2: Yes

5. Review Comments to the Author

Reviewer #1: This is a well-written paper on a very timely and important topic. I appreciated the rationale for undertaking the organizational analysis and agree with the authors' call to action for other networks/organization to undertake a similar activity. In addition to these overall comments, I have the following comments/questions:

1. Given that some NGOs/organizations could consist of many programs, how were such organizations counted and treated in the analysis? For example, I am thinking of an organization such as the Task Force for Global Health, which is an organization itself but also consists of 16-17 programs - many of which have their own mission, objectives, target population of focus, and disease/domains of focus. Each program also participates in different coalitions and has different networks. Was the TFGH counted as one of the 90(ish) organizations in the analysis or were each of its programs counted? Could the authors comment on this and explain how either approach could impact the results?

2. Given the growing interest in making health campaigns, including those campaigns related to NTDs, more collaborative and possibly even include co-delivery of interventions, perhaps the authors would want to discuss how NNN member organizations that address only one NTD might model increased collaboration themselves.

Thank you for the opportunity to review this important paper.

Reviewer #2: I commend the authors for their efforts in conducting this project. Overall, the paper is well written. I have a few comments

1. The authors identify the following "the African region receives nearly twice the support compared to other regions" as a key finding in the analysis, but they do not devote any discussion to this finding. What are the implications of the African region receiving nearly twice the support of other countries? Are other regions being neglected or is the higher proportion of support in the African region backed by data on the concentration of NTDs?

2. The figures in the paper are blurry. Figure 4 in particular is illegible

3. Line 84: the first phrase is redundant with information presented in line 43 in the introduction.

4. Line 178 - numbers less than 10 should be spelled out

5. I recommend editing to ensure consistent formatting of the paragraphs and lists.

6. PLOS authors have the option to publish the peer review history of their article (what does this mean? ). If published, this will include your full peer review and any attached files.

**Do you want your identity to be public for this peer review?** For information about this choice, including consent withdrawal, please see our Privacy Policy .

Reviewer #1: No

Reviewer #2: No

While revising your submission, please upload your figure files to the Preflight Analysis and Conversion Engine (PACE) digital diagnostic tool, https://pacev2.apexcovantage.com/ . PACE helps ensure that figures meet PLOS requirements. To use PACE, you must first register as a user. Registration is free. Then, login and navigate to the UPLOAD tab, where you will find detailed instructions on how to use the tool. If you encounter any issues or have any questions when using PACE, please email PLOS at figures@plos.org. Please note that Supporting Information files do not need this step.

---

## [Editor Report · Decision Letter 1]

2 Dec 2024

The neglected tropical disease non-governmental organization network: the role of a global health network in an evolving global health landscape

PGPH-D-24-01554R1

Dear Dr. Sankar,

We are pleased to inform you that your manuscript 'The neglected tropical disease non-governmental organization network: the role of a global health network in an evolving global health landscape' has been provisionally accepted for publication in PLOS Global Public Health.

Best regards,

Bram Wispelwey, MD, MS, MPH

Academic Editor
